# Defect Engineering Enhances the Charge Separation of CeO_2_ Nanorods toward Photocatalytic Methyl Blue Oxidation

**DOI:** 10.3390/nano10112307

**Published:** 2020-11-21

**Authors:** Jindong Yang, Ning Xie, Jingnan Zhang, Wenjie Fan, Yongchao Huang, Yexiang Tong

**Affiliations:** 1MOE Laboratory of Bioinorganic and Synthetic Chemistry the Key Lab of Low-Carbon Chemistry and Energy Conservation of Guangdong Province, School of Chemistry, Sun Yat-sen University, Guangzhou 510006, China; usayjd@hotmail.com (J.Y.); zhangjn28@mail2.sysu.edu.cn (J.Z.); 2Institute of Environmental Research at Greater Bay Area, Key Laboratory for Water Quality and Conservation of the Pearl River Delta, Ministry of Education, Guangzhou 510006, China; xiening0101@163.com; 3Guangzhou Key Laboratory for Clean Energy and Materials, Guangzhou University, Guangzhou 510006, China; 4Analysis and Testing Center, South China Normal University, Guangzhou 510006, China; 20175033@m.scnu.edu.cn

**Keywords:** cerium oxide, Sm doping, defects, environmental photocatalysis, nanomaterials

## Abstract

Defect-rich photocatalytic materials with excellent charge transfer properties are very popular. Herein, Sm-doped CeO_2_ nanorods were annealed in a N_2_ atmosphere to obtain the defective Sm-doped CeO_2_ photocatalysts (Vo–Sm–CeO_2_). The morphology and structure of Vo–Sm–CeO_2_ were systematically characterized. The Vo–Sm–CeO_2_ nanorods demonstrated an excellent photodegradation performance of methyl blue under visible light irradiation compared to CeO_2_ nanorods and Sm–CeO_2_. Reactive oxygen species including OH, ·O_2_^−^, and h^+^ were confirmed to play a pivotal role in the removal of pollutants via electron spin resonance spectroscopy. Doping Sm enhances the conductivity of CeO_2_ nanorods, benefiting photogenerated electrons being removed from the surface reactive sites, resulting in the superior performance.

## 1. Introduction

Semiconductor-based photocatalytic oxidation reactions have been extensively considered as promising advanced oxidation processes (AOPS) technology for the removal of pollutants in the air and water that have a negative impact on environmental quality, ecosystem safety, and human health [1,2,3,4,5,6,7,8,9,10,11]. However, low photocatalytic efficiency limits its practical application. Thus, various works have been developed to improve its photocatalytic performance via enhanced light absorption and increased the separation efficiency of photogenerated charge carriers [12,13,14,15]. Among them, defect engineering is an efficient method to prepare the ideal photocatalysts [16,17].

Crystallographic defects are generated in materials when the neat arrangement of atoms is broken [18,19,20,21]. At present, defects are mainly prepared by the following methods: hydrogen reduction, calcining under nitrogen atmosphere, strong reducing agent reduction, electric reduction, etc. [22,23,24]. The defects in photocatalysts not only act as recombination centers for free electrons and holes, but also scattering centers for electron and hole travelling, which is not conducive to the diffusion of charge carriers [25,26,27,28]. Defects on the surface of semiconductors can be reactive sites where photoelectrons reduce dissolved O_2_ to O_2_^−^ [29]. Electron trapping sites can consume the photoelectrons, thus preventing the recombination of charge carriers, which can enhance the photocatalytic performance [30,31]. However, the instability of defective materials limits its application because O_2_ gases will refill the defect sites. Therefore, how to use defects effectively to improve the performance and stability of catalysts is one of the research hotspots at present.

Cerium oxide (CeO_2_) has attracted attention in many research fields such as photocatalysis, thermo-catalysis, and electro-catalysis due to its remarkable oxygen-storage ability and redox properties (Ce^4+^/Ce^3+^) [32,33,34,35,36,37]. These properties enhance the release of active oxygen species, balancing the nascent electric charges spontaneously, resulting in defects in CeO_2_ forming and being eliminated quickly [38,39]. Aslam et al. prepared CeO_2-x_ surface defects and used them for the degradation of phenol and its derivatives. Defects serve as the traps and transfer centers to enhance the generation of reactive oxygen species [40]. Jiang and coworkers confirmed the surface-defect dependence of photo-performance [39]. Furthermore, doping a trivalent element into CeO_2_ introduces defects such as Eu-doped CeO_2_ and Yb-CeO_2_ [41,42,43,44]. Our reported work demonstrated that Eu doping can introduce oxygen vacancies into CeO_2_ nanosheets, enhancing the charge transfer. This phenomenon has inspired us to propose enhancing the oxidation and reduction properties and introducing surface defects to improve the photocatalytic performance of CeO_2_.

In this work, we prepared the defective Ce-based photocatalysts (Vo–Sm–CeO_2_) where Sm-doped CeO_2_ nanorods were annealed in a N_2_ atmosphere (Figure 1a). The defects in CeO_2_ were systematically characterized by electron spin resonance (EPR), X-ray photoelectron spectroscopy (XPS), and Raman. The Vo–Sm–CeO_2_ nanorods were tested for the photodegradation of methyl blue (MB) and the results revealed that the photocatalytic activity of Vo–Sm–CeO_2_ was higher than those of pristine CeO_2_ and Sm-doped CeO_2_, which can be attributed to the existence of defects in CeO_2_. Defects in CeO_2_ nanorods enhanced the electrical conduction and promoted charge transfer dynamics. Moreover, the role of defects in producing reactive oxygen species (ROS) was also studied by electron spin resonance spectroscopy.

## 2. Experimental Section

### 2.1. Preparation of Photocatalysts

CeO_2_ was obtained by the following method. Ce(NO_3_)_3_ 6H_2_O was dissolved in distilled water (5 mL). Then, 10 mL of 14 M NaOH was added into the above solution. Finally, the solution was transferred into a Teflon-lined stainless-steel autoclave and it was kept at 110 °C for 24 h. The obtained solid was washed with water and calcined in air at 200 °C for 1 h. Sm doped-CeO_2_ was obtained with the same CeO_2_ nanorods by adding 0.13 g, 0.26 g, and 0.39 g SmCl_3_, respectively. CeO_2_ and Sm–CeO_2_ were annealed in N_2_ gas at 600 °C for 4 h with a ramp rate of 10 °C min^−1^ to obtain Vo–CeO_2_ and Vo–Sm–CeO_2_, respectively. The 2.0 g prepared samples were put into a tube furnace (OTF-1500X-II corundum tube ø 60 mm by Hefei Kejing Materials Technology Co. Ltd., Hefei, China), and the flow rate of N_2_ was continuously pumped into the tube for 30 min at a flow rate of 300 mL min^−1^. The purpose was to drain the air out of the tube and form a high concentration of N_2_ atmosphere in the tube. A flow controller (Beijing Sevenstar Flow Co. Ltd., Beijing, China) was used to maintain the flow stability of N_2_.

### 2.2. Characterization of Photocatalysts

The main instruments used in the experiment are listed as follows: transmission electron microscope (JEM2010-HR, Tokyo, Japan), scanning electron microscope (Gemini SEM 500, Jena, Germany), X-ray diffractometer (D8 ADVANCE, NASDAQ, Billerica, MA, USA), UV−Vis−NIR spectrophotometer (UV-2450, Shimadzu, China), X-ray photoelectron spectroscope (ESCALAB250, Waltham, MA, USA), specific surface area measurements (ASAP 2020V3.03H, Waltham, MA, USA), Raman (Nicolet NXR 9650, Waltham, MA, USA), and a room-temperature photoluminescence spectroscope (FLS920, Edinburgh, UK). The electrochemical tests were carried out with a CHI 660C electrochemical station in a standard three electrode configuration. The illumination source was an AM 1.5 G solar simulator (Newport, LCS 100 94011A (class A), Waltham, MA, USA) directed at the quartz PEC cell (100 mW cm^2^). The working electrode (photoanode) was as follows: 20 mg of the sample was mixed with 2 mL ethyl alcohol to form a slurry and then coated onto a 1 cm × 1 cm fluorine-doped tin oxide (FTO) glass substrate and dried. The reactive species in the photocatalysis were investigated by the electron spin resonance test using the X-band (9.45 GHZ) with 5.00 G modulation amplitude and a magnetic field modulation of 100 kHz. The contact angles of H_2_O drops deposited on the surface of the film were measured at 25 °C using a contact angle meter (SL150, Kino Industrial Co., Ltd., Shanghai, China).

### 2.3. Photocatalytic Performance

In a typical process, 20 mg photocatalyst and 100 mL MB solution (10 mg L^−1^) as a standard pollutant were mixed in a 250 mL reaction vessel with a recirculating cooling water system at 25 °C under simulated solar light irradiation. Prior to the photocatalysis experiment, the sample solution was stirred for 60 min in the dark. The suspension was then exposed to a 300 W xenon lamp light equipped with a UV cutoff filter (λ > 420 nm) under continuous magnetic stirring. At given time intervals, 3 mL suspension from the reaction vessel was pipetted and centrifuged to separate the photocatalyst powder and MB solution. Finally, the absorption spectrum of the supernatant was determined by a UV–Vis spectrophotometer and the absorbance of MB was measured at 665 nm. The degradation efficiency of MB was calculated by the following equation:
Degradation efficiency of MB (%) = (C_0_ − C_t_)/C_0_ × 100%.
where C_0_ and C_t_ represent the initial concentration of MB before irradiation and the residual concentration of MB in solution at irradiation time t, respectively.

## 3. Results and Discussion

### 3.1. The Morphology and Structure Characterization of the Catalysts

The crystal structures of CeO_2_, Sm–CeO_2_, Vo–CeO_2_, and Vo–Sm–CeO_2_ were first identified by X-ray diffraction (XRD). As described in Figure 1b, the peaks of all the prepared samples could be indexed to the (111), (200), (220), (311), (222) planes of the typical cubic structure of CeO_2_ (JCPD#34-0394) [34]. No other new peaks appeared for Sm–CeO_2_ and the Vo–Sm–CeO_2_ samples, suggesting that Sm_2_O_3_ was not produced, while the footprints of Sm could be detected by the X-ray photoelectron spectroscopy (XPS) (Figure 1c). The Sm 3d peaks were located at 1084 eV and 1110 eV, suggesting the existent of Sm^3+^ [45]. Combined with the XRD results, it confirms that Sm was doped into the Sm–CeO_2_ and Vo–Sm–CeO_2_ samples.

The morphology of the prepared samples was observed by scanning electron microscopy (SEM) and transmission electron microscopy (TEM). Appendix A demonstrates the SEM images of CeO_2_, Sm–CeO_2_, Vo–CeO_2_, and Vo–Sm–CeO_2_. Four samples demonstrated almost the same morphology of nanorods, suggesting that doping and introduction of defects did not change the morphology and that the specific surface areas did not have much difference (Appendix A). Moreover, Figure 2a shows the TEM image of Vo–Sm–CeO_2_. Nanorod morphology could be seen and the inset selected area electron diffraction (SAED) pattern demonstrated that it was polycrystalline. High resolution transmission electron microscopy (HRTEM) of pristine CeO_2_ and Vo–Sm–CeO_2_ is provided for contrast (Figure 2b,c). Pristine CeO_2_ demonstrates the well-lined 0.156 nm lattice spacing, which is in accordance with (222) CeO_2_ [34]. Significantly, the inside lattice spacing of Vo–Sm–CeO_2_ became disordered, suggesting that numerous defects are generated after doping Sm. Closer inspection using high-angle annular dark-field scanning TEM (HAADF-TEM) showed the nanorod morphology of Vo–Sm–CeO_2_ (Figure 2d). High multiples of the images of Vo–Sm–CeO_2_ demonstrated that there was an obvious distortion atom, which may originate from Sm doping [46]. Furthermore, the corresponding EDX mapping (Figure 2g) indicated that Ce and Sm were homogeneously distributed in the Vo–Sm–CeO_2_ nanorods.

The existence of defects was confirmed by XPS, Raman, and electron spin resonance spectroscopy [47,48]. Figure 3a shows the O 1s XPS of all samples. Two strong peaks appeared at 529.1 eV and 531.3 eV for pristine CeO_2_ nanorods, which were indexed to lattice oxygen and surface active oxygen [49]. The lattice oxygen peak of Vo–Sm–CeO_2_ shifted 0.3 eV toward low energy due to the effect of Sm doping and the intensity of the surface active oxygen peak increased, suggesting the existence of more surface defects. The surface property of the four samples was detected by the contact angles with the water droplet (Figure 3b). The contact angles decreased after Sm doping and introducing defects, suggesting that Vo–Sm–CeO_2_ was in better contact with water. Furthermore, the Raman spectra of the four samples demonstrated the existence of defects (Figure 3c). The peak at 465 cm^−1^ can be indexed to the vibrational mode of fluorite-type CeO_2_ and the peak at 600 cm^−1^ was attributed to the defects [50]. Figure 3d describes the EPR image of CeO_2_, Sm–CeO_2_, Vo–CeO_2_, and Vo–Sm–CeO_2_. No obvious signal was detected in the pristine CeO_2_ nanorods, while Sm–CeO_2_ and Vo–Sm–CeO_2_ demonstrated a characteristic peak of defect, confirming the approach to creating defective materials [47].

### 3.2. Photocatalytic Performance of the Catalysts

To study the relationship between Sm doping and defects with photocatalytic performance, CeO_2_, Sm–CeO_2_, Vo–CeO_2_, and Vo–Sm–CeO_2_ were used to remove the MB with a 300 W xenon lamp irradiation. Figure 4a shows the photodegradation efficiencies of the four samples. Obviously, the photolysis of MB without photocatalysts can be ignored under our experimental conditions. Pristine CeO_2_ nanorods had 40% photodegradation efficiency after 2 h irradiation. Significantly, Sm doping and defects can enhance the photocatalytic performance of CeO_2_. Sm–CeO_2_ and Vo–CeO_2_ possessed 85% and 80% photodegradation efficiencies, respectively. The Vo–Sm–CeO_2_ sample demonstrated the best degradation performance among the four samples, which could almost entirely remove the MB at 90 min irradiation. The doping amount of Sm was also optimized, which is shown in Appendix A. This result suggests that Sm-doping and surface defects co-promote the photocatalytic activity of CeO_2_. Furthermore, Figure 4b displays the reaction kinetic of photodegradation MB based on Figure 4a, which can be indexed to the Langmuir–Hinshelwood first-order kinetics model. The rate constant value of the Vo–Sm–CeO_2_ sample was 0.012 min^−1^, which was much higher than those of CeO_2_ (0.003 min^−1^), Sm-CeO_2_ (0.007 min^−1^), and Vo-CeO_2_ (0.007 min^−1^). The performance of Vo–Sm–CeO_2_ was also compared with other reported Ce based photocatalysts (Appendix A), suggesting that the Vo–Sm–CeO_2_ sample had a superior photocatalytic performance. Furthermore, the total organic carbon (TOC) removal was performed to identify that the MB removal could be attributed to mineralization. The TOC removal efficiencies (120 min) for MB of CeO_2_, Vo–CeO_2_, Sm–CeO_2_, and Vo–Sm–CeO_2_ samples were 35%, 79%, 75%, and 98%, respectively (Appendix A). This result revealed that most of the MB were mineralized to H_2_O and CO_2_ during our degradation condition.

Photocatalytic stability is an important factor for the application of photocatalysts [51]. Figure 4c depicts the cycling stability of Vo–Sm–CeO_2_ nanorods under visible light irradiation. After five cycles of testing, the photodegradation efficiency of Vo–Sm–CeO_2_ nanorods reduced to 85% and the morphology and the crystal structures remained the same (Appendix A). Defects on the Vo–Sm–CeO_2_ surface of the nanorods can be refilled with oxygen gas, which then affects its stability, as can be observed in other reports. Therefore, we used Vo–Sm–CeO_2_ nanorods to anneal in air and test its performance. Just as we expected, the performance decreased, similar to the performance of the Sm–CeO_2_ nanorods. Interestingly, the degradation efficiency could reach 99% after re-calcining in nitrogen, suggesting that defects can also regenerate on the surface of the Vo–Sm–CeO_2_ nanorods (Figure 4d).

### 3.3. Active Species Trapping Experiments

Active species such as superoxide radicals, hydroxyl radicals, and h^+^ play an important role in advanced oxidation processes (AOPS) technology [52]. To understand which active species generated during the photocatalytic oxidation reaction, we performed active species trapping experiments and electron spin resonance (ESR) measurements [4,34]. Figure 5a shows the photodegradation efficiency of Vo–Sm–CeO_2_ nanorods with different scavengers after 90 min irradiation with visible light (benzoquinone for O_2_^−^, ter-butyl alcohol for OH, and methanol for h^+^). Obviously, the photodegradation efficiencies decreased after adding three scavengers, suggesting that O_2_^−^, OH, and h^+^ are generated during photocatalysis. The active species have a high oxidizing ability to degrade MB into small molecules. Furthermore, Figure 5b,c displays the ESR results of DMPO-·OH and DMPO-O_2_^−^ for Vo–Sm–CeO_2_ nanorods using 5,5-dimethyl-1-pyrriline noxide (DMPO) as a spin trap. No signal corresponding to DMPO-·OH and DMPO-·O_2_^−^ were detected for Vo–Sm–CeO_2_ nanorods in the dark, suggesting ROS did not generated without light irradiation. Four strong peaks were observed when the light was turned on, which was indexed to DMPO-·OH [4,34] and six peaks corresponding to DMPO-O_2_^−^ appeared, suggesting that ·O_2_^−^ was produced during the photocatalysis [30]. The ESR results were consistent with the active species trapping experiments, suggesting that O_2_^−^, OH, and H^+^ play an extremely important role in photocatalysis.

### 3.4. Charge Transfer Analysis

To study the effect of defects and Sm doping in CeO_2_ nanorods, several characterizations were performed including transient photocurrent responses and electrochemical impedance spectra (ZIS) [53,54,55]. Figure 6a shows the transient photocurrent responses of the four samples. The current density value of Vo–Sm–CeO_2_ was higher than those of CeO_2_, Sm–CeO_2_, and Vo–CeO_2_ in the same window, suggesting that Vo–Sm–CeO_2_ had the fast charge transfer. Note that the values of Vo–CeO_2_ and Vo–Sm–CeO_2_ decreased after several ON–OFF cycles, which was in accordance with the stability. Furthermore, the charge behaviors were analyzed by ZIS spectra under visible light irradiation (Figure 6b) [56]. Vo–Sm–CeO_2_ displayed the smallest dimeter among the four samples, suggesting that it had a small charge transfer resistance [57]. The charge transfer resistance value of Sm–CeO_2_ was smaller than that of Vo–CeO_2_, suggesting that Sm doping mainly improved the conductivity of CeO_2_ [58]. Therefore, Sm doping and defects can improve the charge transfer of CeO_2_.

The light absorption range of CeO_2_, Sm–CeO_2_, Vo–CeO_2_, and Vo–Sm–CeO_2_ was observed by UV-Vis spectra (Figure 6c). It can be seen that pristine CeO_2_ nanorods had an absorption region at 420 nm, suggesting that it responds to UV light. After Sm doping and introducing defects into CeO_2_, the absorption band gap displayed little blue shift, suggesting that light absorption was not the main effect for the photocatalytic performance. The bandgaps of four samples could be calculated based on the following formula: *a = A(hv − Eg)^2^/hv* (a is the absorption coefficient and A is the absorption constant for indirect transition) [59,60]. Therefore, the bandgap values of CeO_2_, Sm–CeO_2_, Vo–CeO_2_, and Vo–Sm–CeO_2_ were 2.55 eV, 2.48 eV. 2.75 eV, and 2.51 eV, respectively. Interestingly, Sm doping could narrow the bandgap while defects increased the bandgap [61,62,63]. The relative valence band maximum (VBM) value could be obtained by the XPS valence spectra (Figure 6e). The VBM values of pristine CeO_2_ and Vo–Sm–CeO_2_ were about 1.93 and 1.56 eV, respectively. The schematic for the proposed mechanism is shown in Figure 6f. Doping Sm into the CeO_2_ nanorods changed the location of VBM and CBM, and enhanced the electrical conductivity. Photoelectrons in the CBM of Vo–Sm–CeO_2_ are more conductive to producing reactive oxygen species due to its high potential of CBM. Furthermore, defects on the surface of the Vo–Sm–CeO_2_ nanorods can act as reactor sites for O_2_ reduction. All these results enable Vo–Sm–CeO_2_ nanorods to have better photocatalytic performance.

## 4. Conclusions

In this work, we report on a defective Sm–CeO_2_ nanorod photocatalyst that had a superior photodegradation performance of MB (almost 100%, 90 min) under visible light irradiation. Such performance was achieved due to the synergistic effect of defects and Sm doping, which enhanced the separation of the photogenerated holes and electrons. Sm doping can effectively improve the conductivity of CeO_2_ nanorods and the surface defects can act as reactive sites for photogenerated electrons to reduce O_2_ into ·O_2_^−^. This work not only provides a better understanding of the photocatalytic mechanism, but also offers some guidance for designing a Ce based photocatalyst with high efficient performance.

## Figures and Tables

**Figure 1 nanomaterials-10-02307-f001:**
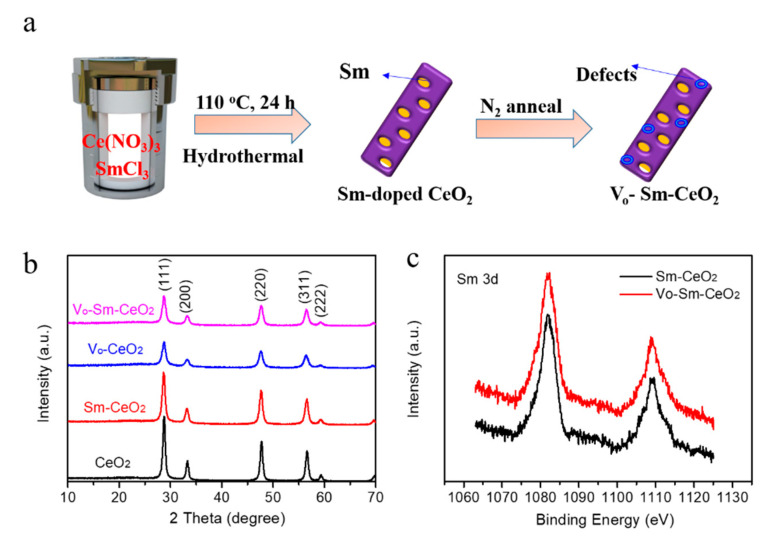
(**a**) Sketch map for preparing Vo–Sm–CeO_2_ sample. (**b**) XRD spectra of CeO_2_, Sm–CeO_2_, Vo–CeO_2_ and Vo–Sm–CeO_2_. (**c**) Sm 3d XPS spectra of Sm–CeO_2_ and Vo–Sm–CeO_2_.

**Figure 2 nanomaterials-10-02307-f002:**
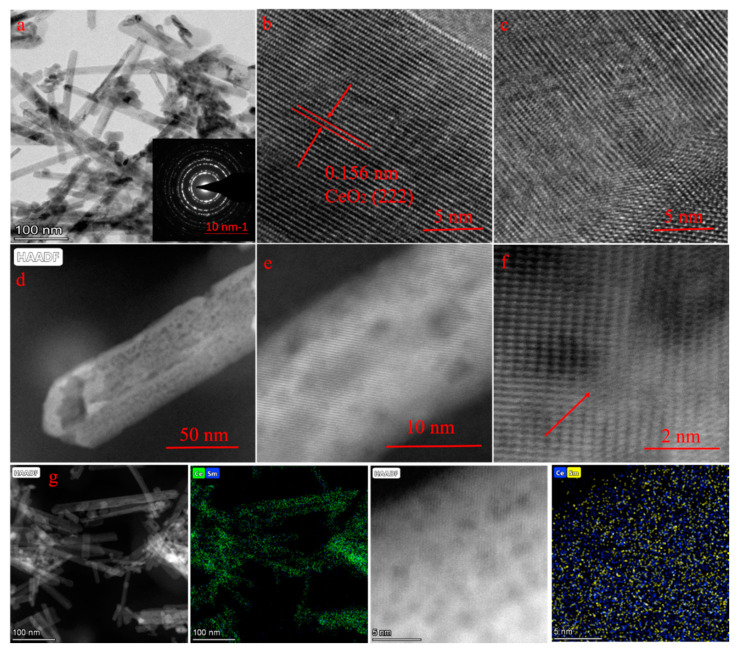
(**a**) Transmission electron microscope (TEM) images of Vo–Sm–CeO_2_, high resolution transmission electron microscopy (HR-TEM) of (**b**) CeO_2_, and (**c**) Vo–Sm–CeO_2_, (**d**–**f**) high-angle annular dark-field scanning TEM (HAADF-STEM) mages of Vo–Sm–CeO_2_, (**g**) energy dispersive X-ray spectroscopy (EDS) images of Vo–Sm–CeO_2_.

**Figure 3 nanomaterials-10-02307-f003:**
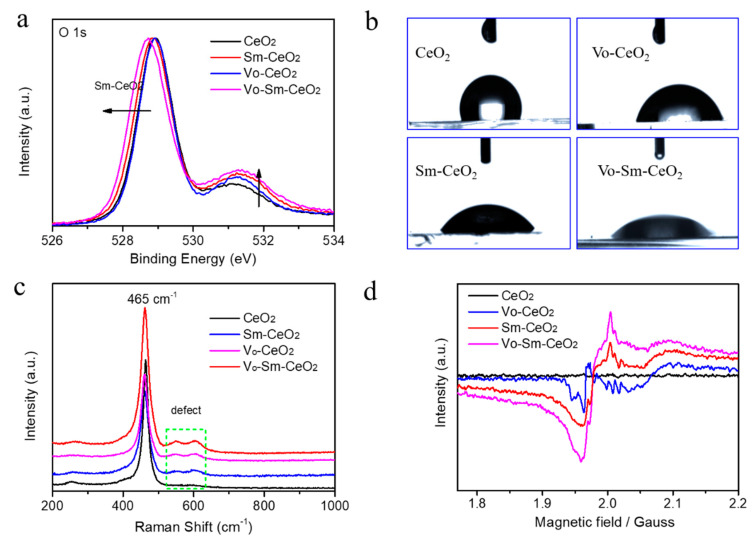
(**a**) O 1s XPS spectra. (**b**) Contact angles with water droplet, (**c**) Raman spectra, and (**d**) EPR spectra of CeO_2_, Sm–CeO_2_, Vo–CeO_2_, and Vo–Sm–CeO_2_.

**Figure 4 nanomaterials-10-02307-f004:**
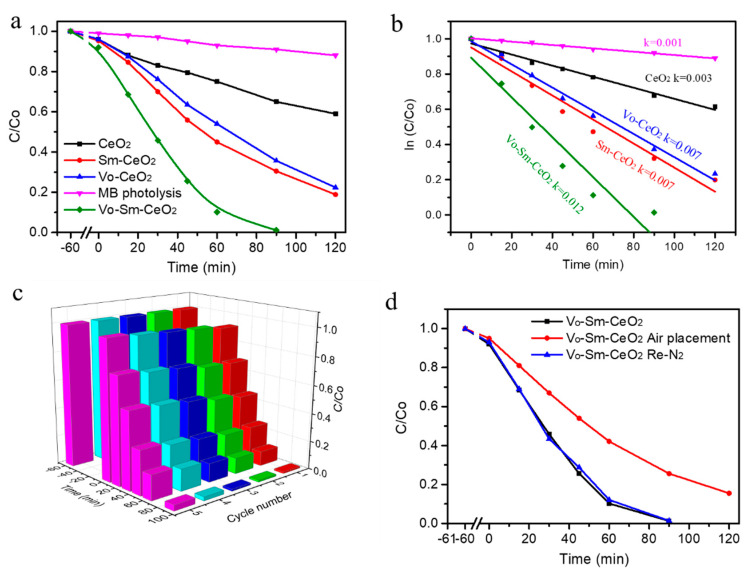
(**a**) Photocatalytic performance of the degradation of MB, (**b**) reaction kinetic, (**c**) stability test toward CeO_2_, Sm–CeO_2_, Vo–CeO_2_, and Vo–Sm–CeO_2_. (**d**) Photocatalytic performance of Vo-Sm-CeO_2_ after treatment.

**Figure 5 nanomaterials-10-02307-f005:**
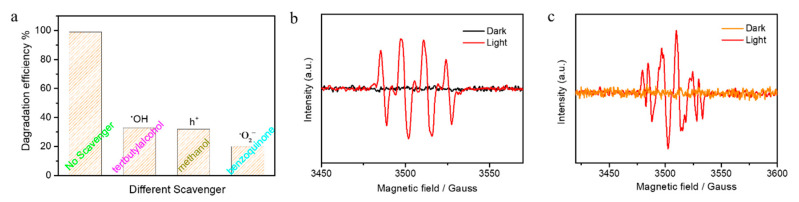
(**a**) Photodegradation efficiency of Vo–Sm–CeO_2_ nanorods with different scavengers after 90 min. The ESR results of the Vo–Sm–CeO_2_ nanorods: (**b**) DMPO-OH and (**c**) DMPO-O_2_^−^.

**Figure 6 nanomaterials-10-02307-f006:**
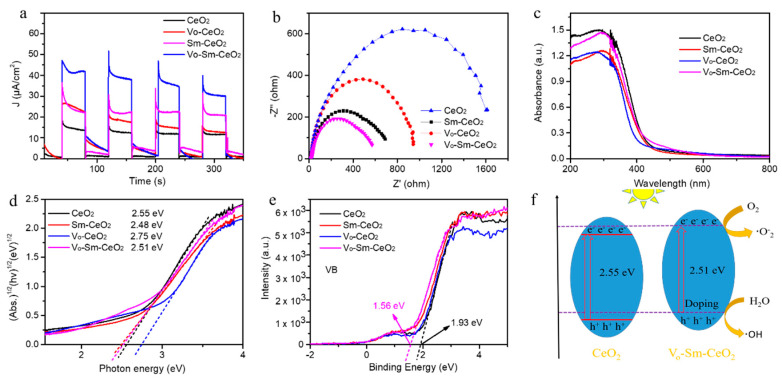
(**a**) Transient photocurrent responses, (**b**) electrochemical impedance spectra (applied potential 0.1 eV), (**c**) UV–Vis spectra, (**d**) optical band gaps, (**e**) VB spectra from XPS of CeO_2_, Sm–CeO_2_, Vo–CeO_2_, and Vo–Sm–CeO_2_. (**f**) Schematic for the proposed mechanism.

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
