# Peer review of "Defect Engineering Enhances the Charge Separation of CeO2 Nanorods toward Photocatalytic Methyl Blue Oxidation"

_nanomaterials, 2020, doi:10.3390/nano10112307_

Round 1

Reviewer 1 Report

This manuscript authored by Yang et al, reported a synergistic effect of the oxygen vacancies (Vo) formation and doping in CeO2 nanorods photocatalyst for the organic pollutant decomposition. The authors prepared CeO2, Sm-CeO2, Vo-CeO2, and Vo-Sm-CeO2 nanorods systematically via a hydrothermal method and followed by additional treatments. Besides, the authors characterized/analyzed the Sm doping and formation of Vo clearly. Importantly, all the measurements and characterizations are comprehensive and fully support the conclusion. Therefore, I recommend this manuscript should be accepted after addressing all the comments/questions/corrections.

1. (Introduction) More comprehensive review/discussion on previous works should be given, i.e., CeO2 nanorods, doping, and formation of Vo in CeO2, the photocatalytic activity of CeO2 and defect engineered CeO2, etc.

2. (experimental section) the preparation of CeO2 nanorods described in the experimental section (synthesis temperature = 110oC) is not consistent with the Figure 1 (temperature = 180oC). Also, a more detailed procedure should be described (washing, the flow rate of N2, type of furnace, amount of powders, etc).

3. (experimental section) the authors used three different doses of SmCl3 for doping. However, only one condition was described in the main text. Add more data or delete the other two conditions.

4. (experimental section) More complete description of all the characterizations of XPS, Raman, contact angle measurement, ESR, J-V, EIS, and UV-Vis should be provided.

5. (Supplementary information.) There is a duplicated experimental section.

6. (Figure 3 and Figure 6) The preparation method of thin-film samples should be described in the experimental section. How’s the film thickness?

7. (Figure 6b) Applied potential should be specified for the EIS spectra. (Figure 6c) Is it from the powder or film samples?

8. All the typos and English checks should be conducted throughout the whole manuscript.

Author Response

Reviewer #1:

Comments:
This manuscript authored by Yang et al, reported a synergistic effect of the oxygen vacancies (Vo) formation and doping in CeO2 nanorods photocatalyst for the organic pollutant decomposition. The authors prepared CeO2, Sm-CeO2, Vo-CeO2, and Vo-Sm-CeO2 nanorods systematically via a hydrothermal method and followed by additional treatments. Besides, the authors characterized/analyzed the Sm doping and formation of Vo clearly. Importantly, all the measurements and characterizations are comprehensive and fully support the conclusion. Therefore, I recommend this manuscript should be accepted after addressing all the comments/questions/corrections.

Response: Thank you for your good comments. We have carefully revised our manuscript thoroughly according to your suggestions. Please see details as follows

1. (Introduction) More comprehensive review/discussion on previous works should be given, i.e., CeO2 nanorods, doping, and formation of Vo in CeO2, the photocatalytic activity of CeO2 and defect engineered CeO2, etc.

Response: Thank you for your good suggestion. More comprehensive review/discussion on previous works are given in the introduction. Please see details in our manuscript (page 2) or as follows:

Aslam et al. prepared CeO2-x surface defects and used for degradation of phenol and its derivatives. Defects serves as the trap and transfer centers to enhance generation of reactive oxygen species [40]. Jiang and his coworkers confirmed the surface-defect dependence of photo-performance [39]. Furthermore, doping trivalent element into CeO2 introduces defects, such as Eu-doped CeO2, Yb-CeO2 [41-44]. Our reported work demonstrated that Eu doping can introduce oxygen vacancies into CeO2 nanosheets, enhancing the charges transfer.

2. (experimental section) the preparation of CeO2 nanorods described in the experimental section (synthesis temperature = 110 oC) is not consistent with the Figure 1 (temperature = 180 oC). Also, a more detailed procedure should be described (washing, the flow rate of N2, type of furnace, amount of powders, etc).

Response: Thank you for your good suggestion. The temperature of preparation of CeO2 nanorods is 110 oC. And the detailed procedure experimental section is described in the revised manuscript. Please see details in our manuscript (page 2) or as follows:

2.0 g prepared samples were put into a tube furnace (OTF-1500X-II corundum tube ø60mm by HEFEI KEJING MATERIALS TECHNOLOGY Co., LTD.), and the flow rate of N2 was continuously pumped into the tube for 30 min at a flow rate of 300 mL/min. The purpose was to drain the air out of the tube and form a high concentration of N2 atmosphere in the tube. The flow controller (Beijing Sevenstar Flow Co.,Ltd) is used to maintain the flow stability of N2.

3. (experimental section) the authors used three different doses of SmCl3 for doping. However, only one condition was described in the main text. Add more data or delete the other two conditions.

Response: Thank you for your good suggestion. The doping amount of Sm was also optimized, which is shown in Figure S3. Sm0.26 –CeO2 has the best performance for photodegradation. So we used Sm0.26 –CeO2 to produce the oxygen vacancies.

Figure S3. Photocatalytic performance of Sm doping CeO2.

4. (experimental section) More complete description of all the characterizations of XPS, Raman, contact angle measurement, ESR, J-V, EIS, and UV-Vis should be provided.

Response: Thank you for your good suggestion. More complete description of all the characterization is also provided. Please see details in our manuscript (page 2) or as follows:

The main instrument used in the experiment: transmission electron microscope (JEM2010-HR), scanning electron microscope (Gemini SEM 500), X-ray diffractometer (D8 ADVANCE), UV−vis−NIR spectrophotometer (UV−vis−NIR, Shimadzu UV-2450), X-ray photoelectron spectroscope (ESCALAB250), specific surface area measurements (ASAP 2020V3.03H), Raman (Nicolet NXR 9650) and room-temperature photoluminescence spectroscope (FLS920, Edinburgh). The electrochemical tests were carried out with a CHI 660C electrochemical station in a standard three electrode configuration. The illumination source was an AM 1.5 G solar simulator [Newport, LCS 100 94011A (class A)] directed at the quartz PEC cell (100 mW cm2). The working electrode (photoanode) as follows: 20 mg of sample was mixed with 2 mL ethyl alcohol to form a slurry and then coated onto a 1 cm *1 cm Fluorine-doped tin oxide (FTO) glass substrate and dried. The reactive species in photocatalysis were investigated by the electron spin resonance test with ESR-DMPO, using the X-band (9.45 GHZ) with 5.00 G modulation amplitude and a magnetic field modulation of 100 kHz. The contact angles of H2O drops deposited on the surface of film were measured at 25 â—¦C using a contact angle meter (SL150, Kino IndustrialCo., Ltd., USA).

5. (Supplementary information.) There is a duplicated experimental section.

Response: Thank you for your good suggestion. The experimental section in supplementary information is removed.

6. (Figure 3 and Figure 6) The preparation method of thin-film samples should be described in the experimental section. How’s the film thickness?

Response: Thank you for your good suggestion. The preparation method of thin-film samples is described in the experimental section. The working electrode (photoanode) as follows: 20 mg of sample was mixed with 2 mL ethyl alcohol to form a slurry and then coated onto a 1 cm *1 cm fluorine-doped tin oxide (FTO) glass substrate and dried.

7. (Figure 6b) Applied potential should be specified for the EIS spectra. (Figure 6c) Is it from the powder or film samples?

Response: Thank you for your good suggestion. Applied potential for the EIS spectra is about 0.1 eV. And is it from film samples.

8. All the typos and English checks should be conducted throughout the whole manuscript.

Response: Thank you for your good suggestion. The English of the text had been fixed. Detailed seen in our revised manuscript.

Reviewer 2 Report

The authors report the photocatalytic performance of an interesting material, however, the presentation seems to be very careless. The following issues need to be addressed before taking decision upon the manuscript.

  1. The ‘characterization techniques’ are fundamental to the research work presented. Hence, they should be the part of the main manuscript rather than the supplementary information. Methods for sample preparation and instrumental settings need to be reported in detail in the main manuscript. Furthermore, the instrumentation of TOC, TPR, ESR and ZIS used in the manuscript also need to be mentioned in this section. Please provide details how contact angle was measured. The ‘photocatalytic performance’ (section 2.3, main manuscript) has been duplicated in the Supplementary Information. What can be the reason behind this?
  2. The acronyms and abbreviations used in the manuscript should be defined at their first use (except for the standard abbreviations). Conversely, there are some abbreviations (such as Vo in abstract and some others in remaining part) in the manuscript which need proper definition/meaning at their first use. The name ‘Raman’ (Page2, line 54) is incomplete to represent the intended technique.
  3. The title of the manuscript seems too broad. Degradation of at least two additional dyes should be added for such generalization.
  4. Since the central theme of the manuscript is the dyes removal, it will be better to introduce organic pollutants and dyes in the beginning part of introduction. I suggest authors to add a couple of sentences or a short paragraph addressing the issue. Two references have been advised for the convenience: Catalysts2019, 9(6), 498; https://doi.org/10.3390/catal9060498; Ceramics International45(11), 13628-13636, https://doi.org/10.1016/j.ceramint.2019.03.239.
  5. Figures; almost every figure has an error or a mistake. Figure 1a; font size of words is bigger (bold also) compared to other figures; it is necessary to maintain consistency in all figures. Figure 1 c; figure consists of spectra of two samples only but the corresponding caption indicates all four samples. Please include the spectra of all four samples. Figure 3b, the font size inside the images is too big compared to neighboring figures. Figure 3d, since ‘a. u.’ has been used in units, the numerical value of intensity is useless. Figure 4a, please write ‘Blank test’ or ‘Photolysis’ instead of ‘MB’ in the legend. Figure 4b, what is the reason for expressing rate constant value with negative sign? Figure 4c, legend/notation is necessary either in figure or in its caption. Figure 5a, please use legend as it is in other figures. Figure 5 b and c, please revise as in Figure 3d (revise other figures wherever applicable). Figure 6c, there is no peak at 500 nm wavelength value, but it is claimed so in line 217. What can be the reason behind this?
  6. Additionally, in Figure 3, I suggest authors to add a figure deconvoluting the O-1s spectra of Vo-Sm-CeO2. Oxygen vacancy as claimed in the manuscript can be further asserted via deconvolution of O-1s spectra.
  7. Page 2, line 62, was it tap water or distilled water? Line 65/66; the sentence is incomplete. Line 84; please express the equation more accurately. Page 3, line 93; it is not necessary to mention ‘JCPD card number’ since a standard reference has been cited. I suggest the authors to rethink in this issue. Line 96, how was it possible to get peaks of Sm 3d5/2 only at two different binding energy values (1084 eV and 1110 eV)?
  8. There are some claims/explanations in the manuscript that need experimental evidence or proper citations. For instance, experimental evidence or proper citations are essential for the claims in lines 111/112, 174, and 213/214.
  9. Manuscript contains several language related mistakes and errors. For instance, third sentence of abstract (lines 17/18) is grammatically incorrect, the intended meaning of the fourth line is not clear, the dots should be used as superscript to represent radicals in the fifth line (similar errors in other parts also), and the sixth sentence is unable to tell the intended meaning. The beginning sentence of the introduction is grammatically wrong and so on. Hence, English language editing of the manuscript is recommended.

Author Response

Reviewer #2:

Comments:

The authors report the photocatalytic performance of an interesting material, however, the presentation seems to be very careless. The following issues need to be addressed before taking decision upon the manuscript.

1. The ‘characterization techniques’ are fundamental to the research work presented. Hence, they should be the part of the main manuscript rather than the supplementary information. Methods for sample preparation and instrumental settings need to be reported in detail in the main manuscript. Furthermore, the instrumentation of TOC, TPR, ESR and ZIS used in the manuscript also need to be mentioned in this section. Please provide details how contact angle was measured. The ‘photocatalytic performance’ (section 2.3, main manuscript) has been duplicated in the Supplementary Information. What can be the reason behind this?

Response: Thank you for your good suggestion. More complete description of all the characterization is also provided. Please see details in our manuscript (page 2) or as follows:

The main instrument used in the experiment: transmission electron microscope (JEM2010-HR), scanning electron microscope (Gemini SEM 500), X-ray diffractometer (D8 ADVANCE), UV−vis−NIR spectrophotometer (UV−vis−NIR, Shimadzu UV-2450), X-ray photoelectron spectroscope (ESCALAB250), specific surface area measurements (ASAP 2020V3.03H), Raman (Nicolet NXR 9650) and room-temperature photoluminescence spectroscope (FLS920, Edinburgh). The electrochemical tests were carried out with a CHI 660C electrochemical station in a standard three electrode configuration. The illumination source was an AM 1.5 G solar simulator [Newport, LCS 100 94011A (class A)] directed at the quartz PEC cell (100 mW cm2). The working electrode (photoanode) as follows: 20 mg of sample was mixed with 2 mL ethyl alcohol to form a slurry and then coated onto a 1 cm *1 cm Fluorine-doped tin oxide (FTO) glass substrate and dried. The reactive species in photocatalysis were investigated by the electron spin resonance test with ESR-DMPO, using the X-band (9.45 GHZ) with 5.00 G modulation amplitude and a magnetic field modulation of 100 kHz. The contact angles of H2O drops deposited on the surface of film were measured at 25 â—¦C using a contact angle meter (SL150, Kino IndustrialCo., Ltd., USA).

2. The acronyms and abbreviations used in the manuscript should be defined at their first use (except for the standard abbreviations). Conversely, there are some abbreviations (such as Vo in abstract and some others in remaining part) in the manuscript which need proper definition/meaning at their first use. The name ‘Raman’ (Page2, line 54) is incomplete to represent the intended technique.

Response: Thank you for your good suggestion. The acronyms and abbreviations used in the manuscript are defined at their first use. Sm-doped CeO2 nanorods are annealed in N2 atmosphere to obtain the defective Sm-doped CeO2 photocatalysts (Vo-Sm-CeO2). Furthermore, the Raman technique is complete represent in the experimental section.

3. The title of the manuscript seems too broad. Degradation of at least two additional dyes should be added for such generalization.

Response: Thank you for your good suggestion. The tittle is revised to “Defect engineering enhances the charge separation of CeO2 nanorods toward photocatalytic methyl blue oxidation”.

4. Since the central theme of the manuscript is the dyes removal, it will be better to introduce organic pollutants and dyes in the beginning part of introduction. I suggest authors to add a couple of sentences or a short paragraph addressing the issue. Two references have been advised for the convenience: Catalysts2019, 9(6), 498; https://doi.org/10.3390/catal9060498; Ceramics International, 45(11), 13628-13636, https://doi.org/10.1016/j.ceramint.2019.03.239.

Response: Thank you for your good suggestion. Semiconductor-based photocatalytic oxidation reactions have been extensive considered as a promising advanced oxidation processes (AOPS) technology for removal pollutants in air and water, which have negative impact on environmental quality, ecosystem safety and human health [1-9].

5. Figures; almost every figure has an error or a mistake. Figure 1a; font size of words is bigger (bold also) compared to other figures; it is necessary to maintain consistency in all figures. Figure 1 c; figure consists of spectra of two samples only but the corresponding caption indicates all four samples. Please include the spectra of all four samples. Figure 3b, the font size inside the images is too big compared to neighboring figures. Figure 3d, since ‘a. u.’ has been used in units, the numerical value of intensity is useless. Figure 4a, please write ‘Blank test’ or ‘Photolysis’ instead of ‘MB’ in the legend. Figure 4b, what is the reason for expressing rate constant value with negative sign? Figure 4c, legend/notation is necessary either in figure or in its caption. Figure 5a, please use legend as it is in other figures. Figure 5 b and c, please revise as in Figure 3d (revise other figures wherever applicable). Figure 6c, there is no peak at 500 nm wavelength value, but it is claimed so in line 217. What can be the reason behind this?

Response: Thank you for your good suggestion. We have revised the Figure based on the reviewers’ suggestion. Please see details in our manuscript.

6. Additionally, in Figure 3, I suggest authors to add a figure deconvoluting the O-1s spectra of Vo-Sm-CeO2. Oxygen vacancy as claimed in the manuscript can be further asserted via deconvolution of O-1s spectra.

Response: Thank you for your good suggestion. Yes, oxygen vacancy can be asserted via deconvolution of O-1s spectra. However, this is not the confirmed evidence of oxygen vacancies. So, we carried out ESR to confirm the existence of oxygen vacancies.

7. Page 2, line 62, was it tap water or distilled water? Line 65/66; the sentence is incomplete. Line 84; please express the equation more accurately. Page 3, line 93; it is not necessary to mention ‘JCPD card number’ since a standard reference has been cited. I suggest the authors to rethink in this issue. Line 96, how was it possible to get peaks of Sm 3d5/2 only at two different binding energy values (1084 eV and 1110 eV)?

Response: Thank you for your good suggestion.

It is distilled water.

The sentence is revised to “Finally, the solution was transferred into Teflon-lined stainless-steel autoclave and it was kept at 110 oC for 24 h.”

The peaks of all the prepared samples can be indexed to (111), (200), (220), (311), (222) planes of the typical cubic structure of CeO2 (JCPD#34-0394)

The Sm 3d peaks are located at 1084 eV and 1110 eV, suggesting the existent of Sm3+, which is also detected in other reported.

8. There are some claims/explanations in the manuscript that need experimental evidence or proper citations. For instance, experimental evidence or proper citations are essential for the claims in lines 111/112, 174, and 213/214.

Response: Thank you for your good suggestion. We have carefully checked the article and some claims/explanations in the manuscript are proper citations. Please see details in our manuscript.

9. Manuscript contains several language related mistakes and errors. For instance, third sentence of abstract (lines 17/18) is grammatically incorrect, the intended meaning of the fourth line is not clear, the dots should be used as superscript to represent radicals in the fifth line (similar errors in other parts also), and the sixth sentence is unable to tell the intended meaning. The beginning sentence of the introduction is grammatically wrong and so on. Hence, English language editing of the manuscript is recommended

Response: Thank you for your good suggestion. The English of the text had been fixed. Detailed seen in our revised manuscript.